# HIVE: HIERARCHICAL VOLUME ENCODING FOR NEURAL IMPLICIT SURFACE RECONSTRUCTION

## ABSTRACT

Neural implicit surface reconstruction has become a new trend in reconstructing a detailed 3D shape from images. In previous methods, however, the 3D scene is only encoded by the MLPs which do not have an explicit 3D structure. To better represent 3D shapes, we introduce a volume encoding to explicitly encode the spatial information. We further design hierarchical volumes to encode the scene structures in multiple scales. The high-resolution volumes capture the high-frequency geometry details since spatially varying features could be learned from different 3D points, while the low-resolution volumes enforce the spatial consistency to keep the shape smooth since adjacent locations possess the same low-resolution feature. In addition, we adopt a sparse structure to reduce the memory consumption at high-resolution volumes, and two regularization terms to enhance results smoothness. This hierarchical volume encoding could be appended to any implicit surface reconstruction method as a plug-and-play module, and can generate a smooth and clean reconstruction with more details. Superior performance is demonstrated in DTU, EPFL, and BlendedMVS datasets with significant improvement on the standard metrics. The code of our method will be made public.

## 1 INTRODUCTION

Surface reconstruction, or image-based modeling (Tan, 2021), from multi-view images is a heavily studied classic task in computer vision. Given multiple images from different views of an object as well as the corresponding camera parameters, this task aims to recover the accurate 3D surface of the target object. Traditional methods (Fuhrmann et al., 2014) usually solve a depth map for each input image and then fuse (Kazhdan et al., 2006) those depth images to build a complete surface model. After the arising of deep networks, many methods try to exploit the neural networks to directly learn the mapping from 2D images to 3D surfaces (Murez et al., 2020; Sun et al., 2021). These learning-based methods skip the intermediate depth map estimation and are highly efficient even on unseen objects and scenes. But they usually only recover coarse scale geometry and miss most of the high frequency surface details.

Recently, many methods (Yariv et al., 2020; Wang et al., 2021; Yariv et al., 2021; Darmon et al., 2022) have achieved highly accurate reconstruction results based on neural implicit surface. These methods represent the 3D shape with an implicit function, such as occupancy (Oechsle et al., 2021)

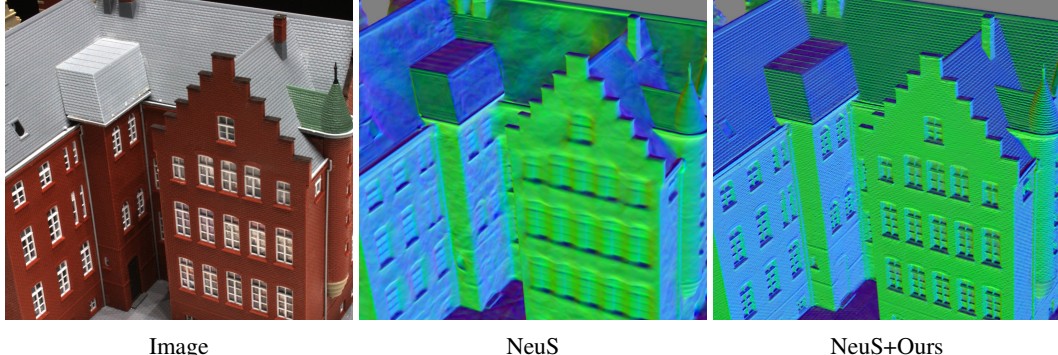

| Image | NeuS | NeuS+Ours |

Figure 1: Visualization of normal maps to highlight our advantages in recovering shape details.

or signed distance fields (SDF) (Wang et al., 2021), and then leverage the neural radiance field (NeRF) (Mildenhall et al., 2020) to render the implicit geometry into color images. Thus, the difference between the rendered images and the input images could optimize the neural radiance field as well as the implicit geometry. Strong results have been demonstrated. However, in these methods, the 3D shape is implicitly encoded in the multi-layer perceptrons (MLPs). Although MLPs are compact and memory efficient, they do not have an explicit 3D structure, which may cause difficulties in optimizing the target 3D shape as is observed in mesh and point-cloud processing (Chibane et al., 2020; Peng et al., 2020). Furthermore, it is also known (Sun et al., 2021) that compact MLPs are hard to encode all the geometry details, such that the recovered surface is prone to be over-smooth.

To solve this problem, some methods employ feature volumes or feature hash tables to facilitate MLPs to encode the 3D space Fridovich-Keil et al. (2022); Sun et al. (2022a); Chen et al. (2022); Müller et al. (2022), which could directly encode the geometry of each 3D position faithfully and unambiguously. However, there are some problems in existing methods. For the volume encoding methods Fridovich-Keil et al. (2022); Sun et al. (2022a); Chen et al. (2022), there is usually only one high-resolution volume in their frameworks, in which case each voxel is updated in isolation. Due to the high degree of freedom in optimizing one voxel, it is hard to maintain a globally coherent shape, as is shown in Figure 2 (c). For the hash-table encoding methods Müller et al. (2022), there are hash collisions which could cause some geometry defects, as is shown in Figure 2 (a) (b). To address these, we introduce a hierarchical volume encoding to encode the 3D space. This hierarchical structure has three advantages. First, the high-resolution feature volume helps to reason high-frequency geometry details in the corresponding locations. Secondly, the voxels in the low-resolution volumes contain the information of large space, which could enforce spatial consistency to keep the shape smooth. Thirdly, this hierarchical structure allows us to use low-dimensional features in the high-resolution volume, which helps to reduce memory consumption. To further reduce memory consumption, we sparsify high-resolution volumes with the preliminary surface reconstruction computed from low-resolution volumes. We only keep voxels nearby the surface of the preliminary results. Finally, we design two smoothness terms to facilitate the optimization to make the reconstructed shape clean.

In the experiments, we demonstrate that by simply adding the proposed hierarchical volume encoding, the performance of different methods are all improved significantly. Specifically, the error of NeuS (Wang et al., 2021) is reduced by 25% from $0.84$ mm to $0.63$ mm, the error of VolSDF (Yariv et al., 2021) is reduced by 23% from $0.86$ mm to $0.66$ mm, and the error of NeuralWarp (Darmon et al., 2022) is reduced by 10% from $0.68$ mm to $0.61$ mm on the DTU (Jensen et al., 2014) dataset. More than that, the error of NeuralWarp is reduced by 31% on the EPFL (Strecha et al., 2008) dataset with the "full" metric. Figure 1 shows an example from the DTU dataset. The color coded normal map clearly demonstrates our method can significantly boost NeuS (Wang et al., 2021) to recover more geometry details while keeping the surface smooth and clean.

The main contributions of this work are summarized in the following:

• We propose a hierarchical volume encoding, which can significantly boost the performance of neural implicit surface reconstruction as a plug-and-play module.

• The hierarchical volume encoding is further improved by employing a sparse structure which reduces the memory consumption, and by enforcing two regularization terms that keep the reconstructed surface smooth and clean.

• We demonstrate superior performance in three datasets.

## 2 RELATED WORK

**Traditional multi-view surface reconstruction** Traditional methods typically follow a long pipeline of structure-from-motion (i.e. camera calibration) (Schönberger & Frahm, 2016), depth map estimation (Barnes et al., 2009), and multi-depth fusion (Kazhdan et al., 2006) to reconstruct surface models from images. Many advanced geometric methods (Jiang et al., 2013; Lhuillier & Quan, 2005; Furukawa & Ponce, 2009; Galliani et al., 2015; Schönberger et al., 2016) have been developed to enhance these different steps in the last two decades.

After the arising of the deep networks, almost all steps in the conventional pipeline have been revolutionized, including feature extraction and matching (DeTone et al., 2018; Sarlin et al., 2020),

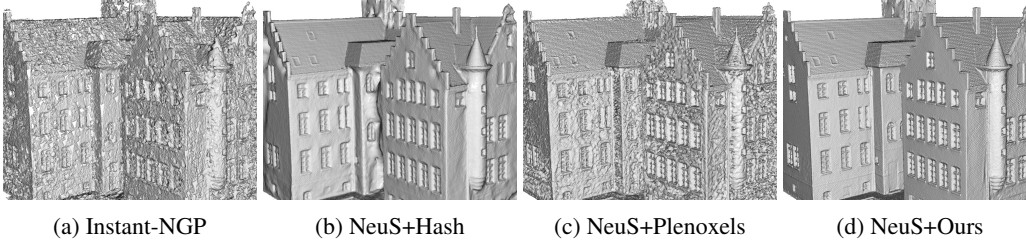

(a) Instant-NGP      (b) NeuS+Hash      (c) NeuS+Plenoxels      (d) NeuS+Ours

Figure 2: Visual comparison of different encoding on DTU scan-24.

structure-from-motion (Ummenhofer et al., 2017; Tang & Tan, 2019), and depth map estimation (Yao et al., 2018; Gu et al., 2020; Tang et al., 2022).

To skip the intermediate depth map estimation, given multiple images with known camera poses, some methods try to directly regress a volumetric prediction end-to-end like an occupancy volume (Ji et al., 2017) or a TSDF volume (Murez et al., 2020; Sun et al., 2021). While these methods simplify the overall pipeline and can be generalized to unseen objects and scenes, they often only learn to recover coarse scale geometry and miss many surface details.

**Implicit surface reconstruction** A neural radiance field (NeRF) encodes a 3D scene (Mildenhall et al., 2020) implicitly in a neural network. The network is optimized to match the rendered images to the input images such that it can generate high quality novel view synthesis results. However, since there is no constraint imposed on the 3D geometry, the surface extracted from the implicit network usually has some defects due to tuning of the density threshold (Oechsle et al., 2021). To address this problem, some methods first represent the 3D shape with an implicit geometry network, like occupancy grid (Niemeyer et al., 2020; Oechsle et al., 2021) or signed distance fields (Yariv et al., 2020; Wang et al., 2021; Yariv et al., 2021; Darmon et al., 2022; Sun et al., 2022b; Long et al., 2022; Fu et al., 2022; Yu et al., 2022), and then transfer the output of the geometry network to a density function, with which the radiance network could render color images. In this way, the radiance network as well as the geometry network can be optimized together by matching the rendered and input images. In these methods, the 3D geometry is directly encoded in the MLPs without any explicit 3D spatial information. To facilitate MLPs, some methods propose to encode the 3D space with a single geometric volume Fridovich-Keil et al. (2022); Sun et al. (2022a); Chen et al. (2022); Martel et al. (2021) or hash tables Müller et al. (2022). However, single-volume may cause noise due to the high degree of freedom in optimizing one voxel, and hash tables may cause hash collisions which leads to defects, as displayed in Figure 2. In this paper, we introduce a hierarchical volume encoding to explicitly encode the 3D spatial information, thus our method can reconstruct more surface details while keeping the shape globally coherent. A similar structure is proposed in Takikawa et al. (2021) for the SDF encoding task, which sums up the features from a large-feature-channel octree, while we concatenate the features from multiple small-feature-channel volumes, in which case our method consumes less memory.

## 3 METHODS

### 3.1 OVERVIEW

The overall framework of our method is illustrated in Figure 3. Given multiple images $\{\mathbf{I}_i\}_{i=1}^{N}$ of an object and corresponding camera poses, the task is to reconstruct the 3D surface of this object. The 3D shape is represented by an implicit SDF network, with which another implicit color network render images using the neural volume rendering. To enhance the representation ability of the spatial information, we add a hierarchical volume encoding as the input of the SDF network, which can embed the 3D space in multiple scales. To save the memory consumption, we also introduce a sparse structure for high-resolution volumes. Finally, two implicit networks as well as the volume encoding are optimized by minimizing the difference between the rendered images and the input images as well as minimizing two regularization terms.

### 3.2 IMPLICIT SURFACE RECONSTRUCTION BASED ON VOLUME RENDERING

Neural implicit volume rendering is first introduced in (Mildenhall et al., 2020) for novel view rendering and then used in (Wang et al., 2021; Yariv et al., 2021; Darmon et al., 2022) for surface reconstruction. Our method could be added to any of them for better reconstruction. Here we take

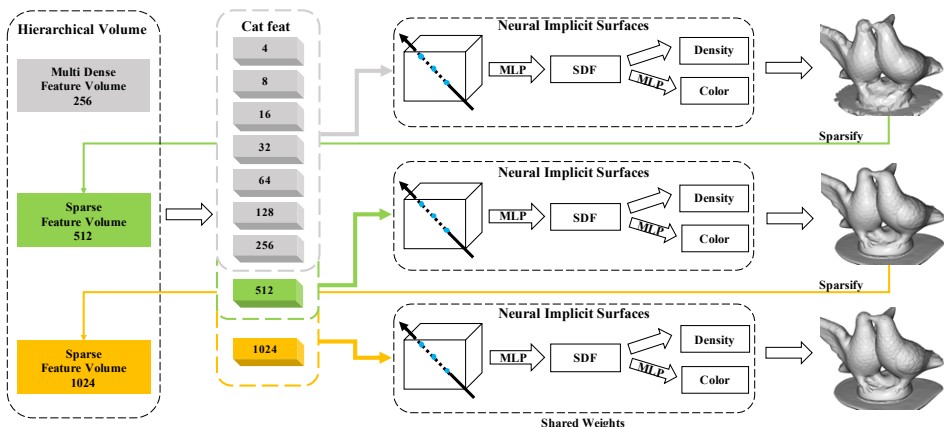

Figure 3: Method overview. In the first stage, we compute an initial result use features from volumes with resolution from 2 to 256. In a later stage, we finalize the result use features from sparsified high resolution volumes with a resolution of 512 or 1024.

NeuS (Wang et al., 2021) as an example, which uses two multi-layer perceptrons (MLPs) to serve as two functions for representing an object: one is the SDF network $sdf : \mathbb{R}^3 \rightarrow \mathbb{R}$ that maps a spatial position $\mathbf{x} \in \mathbb{R}^3$ to its SDF value of the object surface, and the other one is the color network $c : \mathbb{R}^3 \times \mathbb{S}^2 \rightarrow \mathbb{R}$ that maps the spatial point $\mathbf{x}$ and a viewing direction $\mathbf{v} \in \mathbb{S}^2$ to a color value. The surface $\mathcal{S}$ is then represented by the zero-level set of the SDF function as:

$$\mathcal{S} = \{\mathbf{x} \in \mathbb{R}^3 | sdf(\mathbf{x}) = 0\}. \tag{1}$$

To optimize the SDF network, images are rendered from the color network as well as a weighting function computed from the SDF network. The volume rendering function for generating colors is calculated as

$$C(\mathbf{o}, \mathbf{v}) = \int_0^{+\infty} w(t)c(\mathbf{p}(t), \mathbf{v})dt, \tag{2}$$

where $w$ is the weighting function computed by the density from the SDF network.

To make the weighting function unbiased and occlusion-aware, it is calculated as

$$w(t) = T(t)\rho(t), \tag{3}$$

where

$$T(t) = e^{-\int_0^t \rho(u)du}, \tag{4}$$

and

$$\rho(t) = \max(\frac{-\frac{d\Phi_s}{dt}(sdf(\mathbf{p}(t)))}{\Phi_s(sdf(\mathbf{p}(t)))}, 0), \quad \Phi_s(x) = (1 + e^{-sx})^{-1}. \tag{5}$$

### 3.3 HIERARCHICAL VOLUME ENCODING

Previous methods usually encode all the information of a 3D object in the MLPs. To assist the MLPs, we build a 3D feature volume as the input of the MLPs, which explicitly encodes the 3D spatial information. This feature volume can naturally encode the knowledge about the 3D space of the object, while being optimized as well as the MLPs from the rendering loss.

To enhance the representation ability of the encoding, we also employ a hierarchical mechanism, where different scales of the 3D feature volumes are adopted, as is shown in Figure 4. In the experiments, we find a combination of multi-scale volumes works better than a single large-width volume. This is reasonable since in low-resolution volumes, one voxel represent a large space, such that this space can have the same code. The same code could smooth this space and prevent crushing 3D shape, which is important in surface reconstruction.

In a basic setting, we employ eight feature volumes for the 3D space encoding, whose resolutions increase from $2 \times 2 \times 2 \times 4$ to $256 \times 256 \times 256 \times 4$. When a point $\mathbf{x}$ is being rendered, eight features of the corresponding position from these eight volumes are trilinearly interpolated and concatenated to form $\mathcal{F}(\mathbf{x})$, which works as the input of the MLPs. Therefore, the SDF function becomes

$$\mathcal{S} = \{\mathbf{x} \in \mathbb{R}^3 | sdf(\mathcal{F}(\mathbf{x})) = 0\}. \tag{6}$$

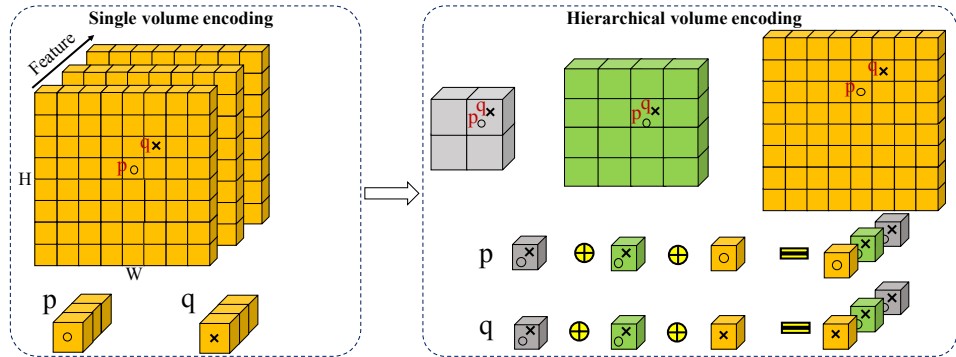

Figure 4: A 2D toy example of the hierarchical volume encoding. Left: a single high resolution volume. Features with 3 channels are used to encode two locations $p$ and $q$. Note it only captures spatial variant features. Right: a hierarchical volume with lower dimensionality. Features have just 1 channel and the memory consumption is much less. The high resolution volume encodes spatial variant features, while the low resolution volume enforces spatial smoothness.

## 3.4 Sparse High Resolution Volume

It is evident that the higher the resolution of the volume is, the more details would be recovered, but a volume larger than $256 \times 256 \times 256$ would consume much more memory. To solve this problem, a multi-stage optimization scheme is adopted with high-resolution sparse volumes. In general, a lot of voxels in a dense volume are unoccupied and invalid. Thus, we could save memory consumption a lot by using the shape reconstructed from the coarse volumes to cull voxels far from the surface in the high-resolution volumes.

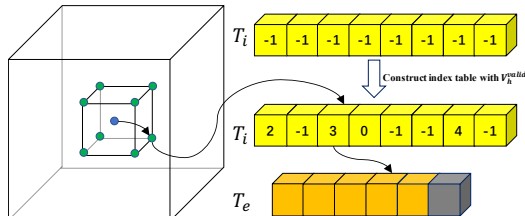

Figure 5: Sparse high-resolution volume. The index of $-1$ fetches the last embedding (in dark gray) in $T_e$.

To do this, we propose a multi-stage optimization framework. In the first stage, we use the above-mentioned basic structure, where the resolution of the largest volume is 256. After the first stage, we obtain a coarse surface reconstruction $\mathcal{S}_c$. In the second stage, we utilize $\mathcal{S}_c$ to cull unnecessary voxels to obtain the valid voxels $\mathbf{V}_h^{valid}$ in the high-resolution volume. Specifically, we first dilate the surface $\mathcal{S}_c$ to ensure all the valid voxels nearing the surface are included. Therefore, we only need to optimize the embeddings of $n$ valid voxels in $\mathbf{V}_h^{valid}$ instead of all voxels. Here, we use a simple data structure, an embedding table $T_e$, to store these embeddings.

Given a floating-point three-dimensional position $\mathbf{x}$, we obtain the surrounding eight integer coordinates through the rounding operation, take out the embedding of these integer points, and then fuse them through trilinear interpolation to serve as the encoding of $\mathbf{x}$. In order to efficiently extract the corresponding embedding from the embedding table $T_e$, we construct another index table $T_i$ which stores the indexes of $T_e$. The values in table $T_i$ are all initialized to $-1$, which corresponds to the last embedding in table $T_e$. The length of the embedding table $T_e$ is only $n+1$ ($n \ll N^3$), such that the memory consumption is reduced a lot due to the sparse structure of the high-resolution volumes. More details are given in the appendix.

## 3.5 Loss Function

We equip three previous methods (Wang et al., 2021; Yariv et al., 2021; Darmon et al., 2022) with our hierarchical volume encoding. To optimize the model, we use the losses in their work without changes, i.e. a rendering loss $\mathcal{L}_{color}$ which minimizes the difference between the rendered colors and input colors, and an Eikonal loss $L_{eik}$ (Gropp et al., 2020) which encourages unit norm of the SDF function gradients. Besides, for NeuralWarp (Darmon et al., 2022), an additional warping color loss $\mathcal{L}_{warp}$ is adopted, which warps views on each other to enforce multi-view photo-consistency.

| Method | 24 | 37 | 40 | 55 | 63 | 65 | 69 | 83 | 97 | 105 | 106 | 110 | 114 | 118 | 122 | Mean |
|---|---|---|---|---|---|---|---|---|---|---|---|---|---|---|---|---|
| IDR(Yariv et al., 2020) | 1.63 | 1.87 | 0.63 | 0.48 | 1.04 | 0.79 | 0.77 | 1.33 | 1.16 | 0.76 | 0.67 | 0.90 | 0.42 | 0.51 | 0.53 | 0.90 |
| MVSDF(Zhang et al., 2021) | 0.83 | 1.76 | 0.88 | 0.44 | 1.11 | 0.90 | 0.75 | 1.26 | 1.02 | 1.35 | 0.87 | 0.84 | 0.34 | 0.47 | 0.46 | 0.88 |
| COLMAP(Schönberger et al., 2016) | 0.45 | 0.91 | 0.37 | 0.37 | 0.90 | 1.00 | 0.54 | 1.22 | 1.08 | 0.64 | 0.48 | 0.59 | 0.32 | 0.45 | 0.43 | 0.65 |
| NeRF(Mildenhall et al., 2020) | 1.90 | 1.60 | 1.85 | 0.58 | 2.28 | 1.27 | 1.47 | 1.67 | 2.05 | 1.07 | 0.88 | 2.53 | 1.06 | 1.15 | 0.96 | 1.49 |
| UNISURF(Oechsle et al., 2021) | 1.32 | 1.36 | 1.72 | 0.44 | 1.35 | 0.79 | 0.80 | 1.49 | 1.37 | 0.89 | 0.59 | 1.47 | 0.46 | 0.59 | 0.62 | 1.02 |
| NeuS(Wang et al., 2021) | 1.00 | 1.37 | 0.93 | 0.43 | 1.10 | 0.65 | 0.57 | 1.48 | 1.09 | 0.83 | 0.52 | 1.20 | 0.35 | 0.49 | 0.54 | 0.84 |
| VolSDF(Yariv et al., 2021) | 1.14 | 1.26 | 0.81 | 0.49 | 1.25 | 0.70 | 0.72 | 1.29 | 1.18 | 0.70 | 0.66 | 1.08 | 0.42 | 0.61 | 0.55 | 0.86 |
| NeuralWarp(Darmon et al., 2022) | 0.49 | 0.71 | 0.38 | 0.38 | 0.79 | 0.81 | 0.82 | 1.20 | 1.06 | 0.68 | 0.66 | 0.74 | 0.41 | 0.63 | 0.51 | 0.68 |
| NeuS+Hash | 1.26 | 1.67 | 0.84 | 0.39 | 1.19 | 0.78 | 0.64 | 1.41 | 1.24 | 0.72 | 0.58 | 1.49 | 0.34 | 0.49 | 0.52 | 0.90 |
| NeuS+Ours | **0.41** | 0.70 | **0.35** | **0.35** | 0.85 | **0.55** | **0.62** | 1.33 | 1.03 | 0.67 | 0.53 | 0.79 | **0.33** | **0.42** | **0.45** | 0.63 |
| VolSDF+Ours | 0.51 | 0.77 | 0.49 | 0.38 | 0.75 | 0.67 | 0.71 | 1.16 | 1.00 | **0.63** | 0.57 | 0.89 | 0.38 | 0.52 | 0.48 | 0.66 |
| NeuralWarp+Ours | 0.44 | **0.69** | 0.36 | 0.38 | **0.63** | 0.64 | 0.69 | **1.13** | **0.99** | 0.66 | **0.51** | **0.61** | 0.36 | 0.53 | 0.52 | **0.61** |

Table 1: Quantitative results on the DTU dataset.

In addition, we add two additional regularization terms $\mathcal{L}_{tv}$ and $\mathcal{L}_{normal}$ to make the reconstructed surfaces smooth and clean. The total variation (TV) (Rudin & Osher, 1994) regularization is applied to each embedding volume to make adjacent voxels have similar characteristics, in which case the geometry could be more continuous and compact. It is computed as:

$$\mathcal{L}_{tv} = \sum_m \sum_{i,j,k} \sqrt{(V_{i+1,j,k} - V_{i,j,k})^2} + \sqrt{(V_{i,j+1,k} - V_{i,j,k})^2} + \sqrt{(V_{i,j,k+1} - V_{i,j,k})^2}, \quad (7)$$

where $m$ is the number of the hierarchical volumes.

Another regularization term $\mathcal{L}_{normal}$ is a smoothness constraint for normal. For each pixel of the image, we calculate the accumulated normal gradients along the marching ray as

$$N_{grad}(\mathbf{o}, \mathbf{v}) = \int_0^{+\infty} w(t) n_{grad}(t) dt, \quad (8)$$

where $n_{grad}(t)$ is the gradient of the normal at $\mathbf{p}(t)$ and computed as

$$n_{grad}(t) = \nabla^2 sdf(\mathcal{F}(\mathbf{p}(t))). \quad (9)$$

The normal regularization $\mathcal{L}_{normal}$ is then defined as:

$$\mathcal{L}_{normal} = \frac{1}{N_{pix}} \sum_{pix} ||N_{grad}||_2, \quad (10)$$

where $N_{pix}$ is the number of pixels in one optimization.

Thus, the final loss $\mathcal{L}$ is defined as:

$$\mathcal{L} = \mathcal{L}_{ori} + \lambda_{tv} * \mathcal{L}_{tv} + \lambda_{normal} * \mathcal{L}_{normal}, \quad (11)$$

where $\mathcal{L}_{ori}$ is the original loss of each method, $\lambda_{tv}$ and $\lambda_{normal}$ are two weighting factors.

## 4 EXPERIMENTS

### 4.1 IMPLEMENTATION DETAILS

This work is implemented in Pytorch and experimented on Nvidia 2080Ti GPUs. The Adam optimizer $(0.9, 0.999)$ is used to update the network weights. The learning rate for the MLPs is set to $5e^{-4}$ and decreased to $1/20$, while the learning rates for the volumes with resolutions of $2, 4, 8, 16, 32, 64, 128, 256, 512, 1024$ are set to $0.01, 0.01, 0.01, 0.01, 0.01, 0.001, 0.001, 1e^{-4}, 1e^{-4}, 1e^{-4}$ and decreased to $1/100$. We adopt a three-stage training strategy by default, and the number of optimization iterations is set to the same as the original framework. Taking "NeuS + Ours" as an example, there are 300K iterations in total. In the first stage, eight volumes of resolution from $[2^3, 4]$ to $[256^3, 4]$ are employed for 80K-iteration optimization. In the second stage, a sparse volume with the resolution of $[512^3, 4]$ is added for another 20K-iteration optimization. Finally, a sparse volume of $[1024^3, 4]$ is appended for the remaining 200K-iteration optimization. The length of the index tables for the second and the third stages are set to $256^3$.

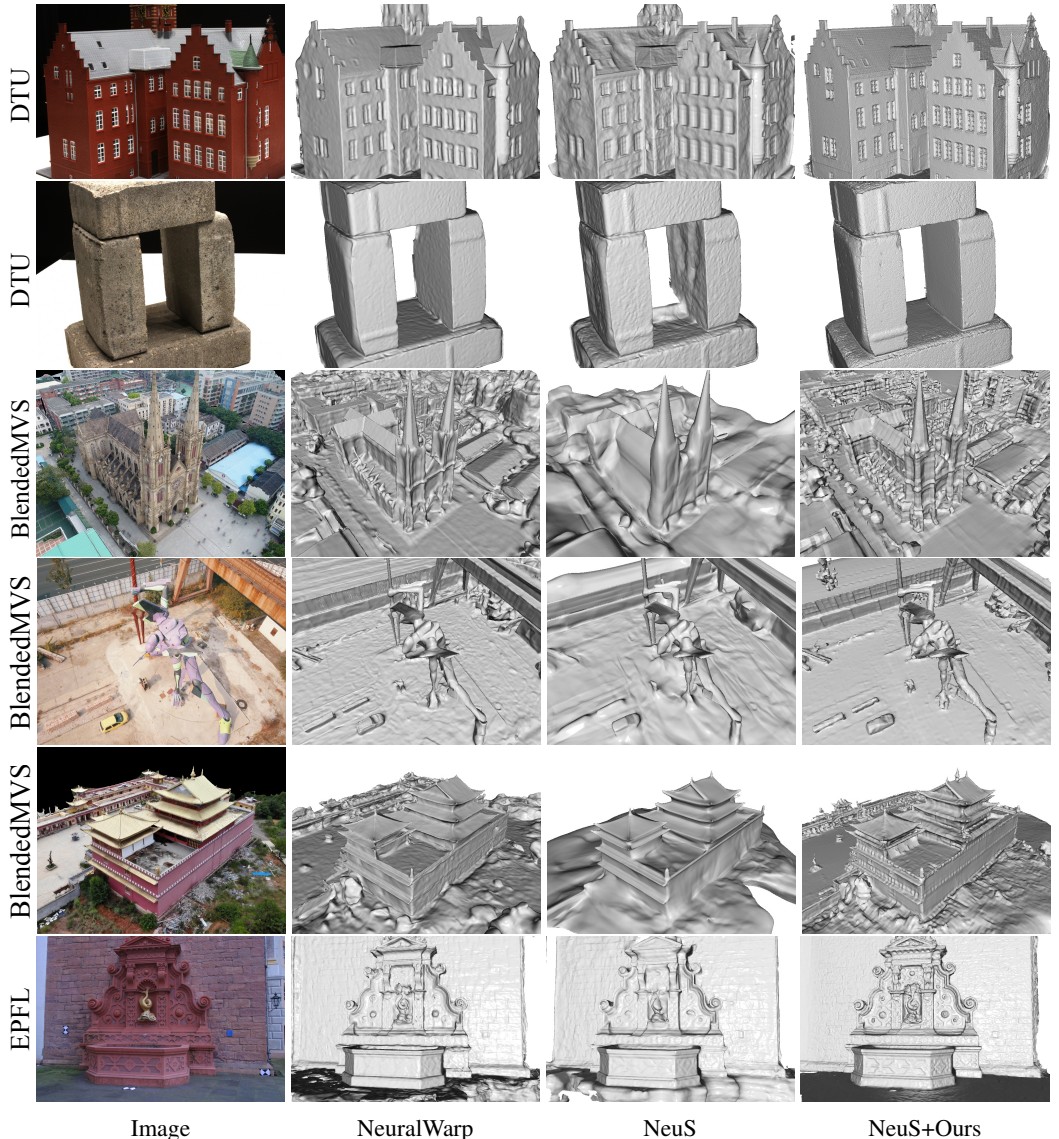

Figure 6: Visual comparison of the reconstructed meshes.

## 4.2 EVALUATION

For a fair comparison, we follow previous methods to evaluate our method on DTU (Jensen et al., 2014) , EPFL (Strecha et al., 2008) and BlendedMVS (Yao et al., 2020) benchmark. The Chamfer L1 distance is used for evaluating the accuracy of the recovered surfaces. This metric is the average of the accuracy, which measures the distance from the reconstructed surface to the ground-truth surface, and the completeness, which measures the distance in reverse.

We first evaluate our method on the DTU dataset and report the quantitative results in Table 1. For a fair comparison, the meshes of all methods are extracted by marching cubes with resolution of 512. We follow previous work (Darmon et al., 2022; Oechsle et al., 2021) to clean the predicted mesh by visibility masks for more reasonable evaluation. As shown in Table 1, the accuracy of previous methods is improved significantly by adding the hierarchical volume encoding. To be specific, the error of NeuS (Wang et al., 2021) is reduced by 25%, from 0.84 to 0.63, while the error of NeuralWarp (Darmon et al., 2022) is reduced by 10%, from 0.68 to 0.61. The reconstructed meshes are shown in Figure 6, from where we can see that the surfaces of our method are smooth and clean, while containing accurate geometry details. For instance, the details of the house in the

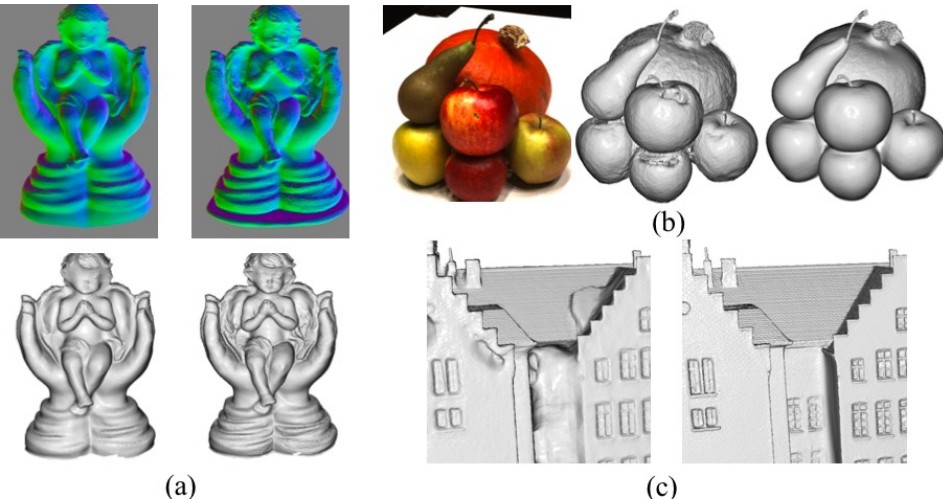

Figure 7: Ablation study. (a) Sparse high-res volume. The results of without and with sparse high-resolution volumes are displayed in the first and second rows, respectively. The top row shows the normal maps and the bottom row shows the reconstructed meshes. (b) Regularization terms. The color image, the results of without and with regularization terms are displayed. (c) NeuS+Hash vs. NeuS+Ours.

first scene in Figure 6 are recovered, especially the shape of the windows and the bricks of the roof, which is more remarkable in the normal map in Figure 1.

We then evaluate our method on the EPFL dataset. For a fair comparison, we follow NeuralWarp to use both the "full" chamfer distance and the "center" chamfer distance to evaluate the reconstructed surfaces. The "center" metric only evaluates the center of the scene cropped by a manually defined box, which focuses more on the precision of the reconstruction, while the "full" metric is also influenced by the ground plane and the rarely seen points, which thus also considers the completeness of the recon-

| Method | Fountain-P11 | | Herzjesu-P7 | | Mean | |
|---|---|---|---|---|---|---|
| | Full | Center | Full | Center | Full | Center |
| COLMAP | 6.47 | 2.45 | **7.95** | 2.31 | 7.21 | 2.38 |
| UNISURF | 26.16 | 17.72 | 27.22 | 13.72 | 26.69 | 15.72 |
| MVSDF | 6.87 | 2.26 | 11.32 | 2.72 | 9.10 | 2.49 |
| VolSDF | 12.89 | 2.99 | 13.61 | 4.58 | 13.25 | 3.78 |
| NeuS | 7.64 | 2.25 | 19.03 | 14.02 | 13.33 | 7.79 |
| NeuralWarp | 7.77 | 1.92 | 8.88 | 2.03 | 8.32 | 1.97 |
| NeuS+Ours | 7.39 | 1.76 | 11.23 | 3.69 | 9.31 | 2.73 |
| NeualWarp+Ours | **3.31** | **1.73** | 8.13 | **1.71** | **5.72** | **1.72** |

Table 2: Quantitative results on the EPFL dataset.

struction. As shown in Table 2, the performance of each framework is improved significantly by adding our method. Especially, the "full" metric of NeuralWarp is decreased by $31\%$ to $5.72$, which is also a $21\%$ improvement compared to COLMAP (Schönberger et al., 2016).

Finally, the evaluation is performed on the BlendedMVS dataset, which contains objects of more complex shapes. As shown in Figure 6, our method can recover more detailed shapes than previous methods. These illustrate that our hierarchical volume encoding can facilitate MLPs to encode more complex shapes and improve the neural implicit surface reconstruction.

## 4.3 ABLATION STUDY

We choose "NeuS + Ours" to perform extensive ablation studies to validate the proposed method. All the results are reported from the 15 scans of DTU dataset unless otherwise specified.

**Hierarchical volume encoding.** To study the effectiveness of the hierarchical volume encoding, we equip the basic set of eight volumes to NeuS (Wang et al., 2021) without any other change. As shown in Table 3, the error of the recovered surfaces is significantly reduced from $0.84$ to $0.70$.

**Regularization terms.** We also ablate the two smoothness terms to study their effectiveness. As shown in Table 3, they help to further reduce the error metric down to $0.66$. Figure 7 (b) shows a visual comparison without (middle) and with (right) the regularization terms. The reconstructed mesh is smoother and more complete with the regularization terms.

**Sparse high resolution volume.** We analyze the effect of the resolution of volumes. As shown in Table 3, sparse high resolution embedding volume (resolution of 1024 here) further improves the accuracy to $0.63$. We also visualize the reconstructed surfaces without (left) and with (right) sparse

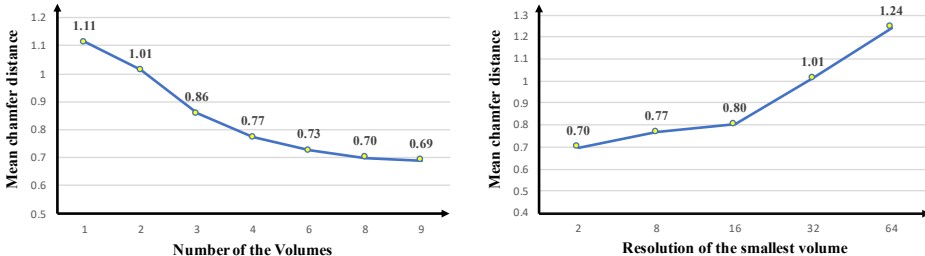

Figure 8: Ablation study about the number of the volumes and the resolution combination of volumes.

high resolution embedding in Figure 7(a). It is clear that the high-resolution embedding generates more shape details, e.g., clearer facial shapes.

**Setup of embedding volumes.** To further inspect the effect of different combinations of the volume encoding, we perform two experiments on the volume setups. (1) We study the number of the volumes. We first compare our hierarchical setup to a large volume of $256^3 \times 32$, the number of whose learnable parameters is about 8 times larger than ours. However, its performance is much worse than ours, as is presented in the left of Figure 8. Then we keep the number of feature channels fixed at 32 and increase the volume number. The resolution of the first volume is set to 256, and the resolution of the remaining volumes is decreased by $1/2$ one by one. As shown in Figure 8 left, the accuracy of the reconstructed surface is gradually improved with the arising of the volume number. (2) Next, we study the impact of the low-resolution volume. We keep the resolution of the largest volume fixed at 256, the number of the volumes fixed at 8, and vary the resolution of the smallest volume from 2 to 64. The resolutions of intermediate volumes are sequentially enlarged, and the enlargement factor is $(256/min\_res)^{1/8}$. From the results shown in Figure 8 right, we can see the performance is decreased when the low-resolution volume begins from a larger resolution. This is intuitive because the low-resolution volume enforces spatial smoothness. Starting from a lower resolution helps to enforce smoothness across larger areas.

| Hierarchical Volume | Regular Terms | Sparse High-Res | Mean |
|:---:|:---:|:---:|:---:|
| ✗ | ✗ | ✗ | 0.84 |
| ✓ | ✗ | ✗ | 0.70 |
| ✓ | ✓ | ✗ | 0.66 |
| ✓ | ✓ | ✓ | 0.63 |

Table 3: Ablation study on the DTU dataset.

**Layer number of MLPs.** Finally, we experiment with the layer number of MLPs in the SDF network to inspect its effect. When only half of the layers are used, the chamfer distance increases to 0.71. More details are included in the appendix.

### 4.4 COMPARISON TO HASH ENCODING

A similar hierarchical feature encoding is adopted in Instant-NGP (Müller et al., 2022), but the features are stored in hash tables such that the hash collision is inevitable. Although this is efficient in both memory and convergence speed, the hash collision may result in defects in the implicit geometry. To make a comparison to hash encoding, we perform two experiments: one is the original version of Instant-NGP, and the other one is adding the hash encoding to NeuS framework. Some of the results are presented in Figure 7(c) and Table 1. For more details and results please see the appendix. From the results, we can see that although Instant-NGP or Hash+NeuS could obtain accurate novel view synthesis, there are some defects in their surface. However, the convergence speed of Instant-NGP is faster than our method, which is due to the fact that we still employ the large MLPs used in NeuS.

### 5 CONCLUSION

We propose to explicitly encode the 3D shape surface by hierarchical volumes to facilitate MLPs in neural implicit surface reconstruction methods. The spatially varying features can be obtained from high resolution volume to reason more details for each query 3D point, while the feature from low-resolution volumes could reason spatial consistency to keep shapes smooth. We also design a sparse structure to reduce the memory consumption of high-resolution volumes, and two regularization terms to enhance surface smoothness. Our hierarchical volume encoding could be appended to any implicit surface reconstruction method as a plug-and-play module to significantly boost their performance, which is demonstrated in three datasets.

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

# A APPENDIX

## A.1 DATASETS

DTU (Jensen et al., 2014) is a well-used dataset for multi-view reconstruction, consisting of 124 scans of various objects. The images are obtained from an RGB camera and a structured light scanner mounted on an industrial robot arm. Each scene is captured from 49 or 64 views with a resolution of $1600 \times 1200$. For a fair comparison, we follow previous methods and select 15 scenes from DTU for evaluation. BlendedMVS (Yao et al., 2020) is another dataset for multi-view reconstruction composed of 113 scenes including architectures, sculptures, and small objects. Each scene consists of dozens to hundreds of images with a resolution of $768 \times 576$. EPFL (Strecha et al., 2008) is a small dataset composed of two outdoor scenes, Fountain and Herzjesu, which contain 11 and 9 high-resolution images, respectively, and the accurate ground truth meshes.

## A.2 DETAILS OF SPARSE VOLUME STORING

We look up the values of $\mathbf{V}_h^{valid}$ from table $T_e$ through $T_i$. Specifically, we first define a mapping function $f$:

$$f(\mathbf{x}) = x + y * N + z * N^2, \tag{12}$$

where $N$ is the resolution of the volume, and $x, y, z$ are the coordinates of the position $\mathbf{x}$. We utilize $f$ to map each voxel $\mathbf{v}_j$ in $\mathbf{V}_h^{valid}$ to the index in table $T_i$, and then $T_i$ further converts it to the index $j$ in $T_e$:

$$T_i(f(\mathbf{x})) = j, \; j = 1...n \tag{13}$$

where $n$ is the number of valid voxels in $\mathbf{V}_h^{valid}$. The length of the embedding table $T_e$ is only $n+1$ ($n \ll N^3$), such that the memory consumption is reduced a lot due to the sparse structure of the high-resolution volumes.

## A.3 LIMITATIONS

(1) Our framework is memory bounded and has difficulties in reconstructing large-scale scenes. This is also a common problem of volume-based methods. (2) The voxels pruned at low resolution cannot be recovered at high resolution. To address this, we dilate the low-resolution surface, such that usually all the useful voxels would be included. But there is still a chance that the voxels would be pruned by mistake.

## A.4 MORE ABLATION STUDY

**Layer number of MLPs** We perform experiments on the layer number of MLPs in the SDF network to inspect its effect. As shown in Table 4, the mean chamfer distance decreases with the increase of the layer number of MLPs. NeuS (Wang et al., 2021) adopts 9 layers of MLPs and gets an error of 0.84, while the error of our method already decreases to 0.78 with only 2 layers.

| Method | Num. | 24 | 37 | 40 | 55 | 63 | 65 | 69 | 83 | 97 | 105 | 106 | 110 | 114 | 118 | 122 | Mean |
|--------|------|----|----|----|----|----|----|----|----|----|-----|-----|-----|-----|-----|-----|------|
| NeuS | 9 | 1.00 | 1.37 | 0.93 | 0.43 | 1.10 | 0.65 | 0.57 | 1.48 | 1.09 | 0.83 | 0.52 | 1.20 | 0.35 | 0.49 | 0.54 | 0.84 |
| NeuS+Ours | 1 | 1.59 | 1.67 | 0.56 | 0.94 | 0.96 | 0.73 | 0.97 | 1.95 | 1.82 | 0.95 | 1.48 | 1.67 | 1.73 | 0.83 | 0.78 | 1.24 |
| | 2 | 0.53 | 0.78 | 0.73 | 0.36 | 0.90 | 0.66 | 0.74 | 1.48 | 1.10 | 0.72 | 0.55 | 1.59 | 0.36 | 0.57 | 0.64 | 0.78 |
| | 3 | 0.49 | 0.75 | 0.72 | 0.37 | 0.87 | 0.58 | 0.63 | 1.34 | 1.11 | 0.74 | 0.49 | 1.58 | 0.34 | 0.42 | 0.46 | 0.73 |
| | 5 | 0.51 | 0.74 | 0.58 | 0.36 | 0.89 | 0.58 | 0.66 | 1.39 | 1.05 | 0.78 | 0.51 | 1.47 | 0.36 | 0.41 | 0.42 | 0.71 |
| | 9 | 0.42 | 0.77 | 0.38 | 0.34 | 0.81 | 0.61 | 0.65 | 1.26 | 1.10 | 0.65 | 0.49 | 1.60 | 0.32 | 0.58 | 0.51 | 0.70 |

Table 4: Quantitative results for the different layer number of MLPs on the DTU dataset.

**Ablation study on NeuralWarp+Ours** To better inspect the effectiveness of our modules on different frameworks, we present the ablation study on the NeuralWarp Darmon et al. (2022) framework, as shown in Table 5 and Figure 9.

| Hierarchical Volume | Regular Terms | Sparse High-Res | Mean |
|:---:|:---:|:---:|:---:|
| ✗ | ✗ | ✗ | 0.68 |
| ✓ | ✗ | ✗ | 0.64 |
| ✓ | ✓ | ✗ | 0.63 |
| ✓ | ✓ | ✓ | 0.61 |

Table 5: Ablation study of NeuralWarp+Ours on the DTU dataset.

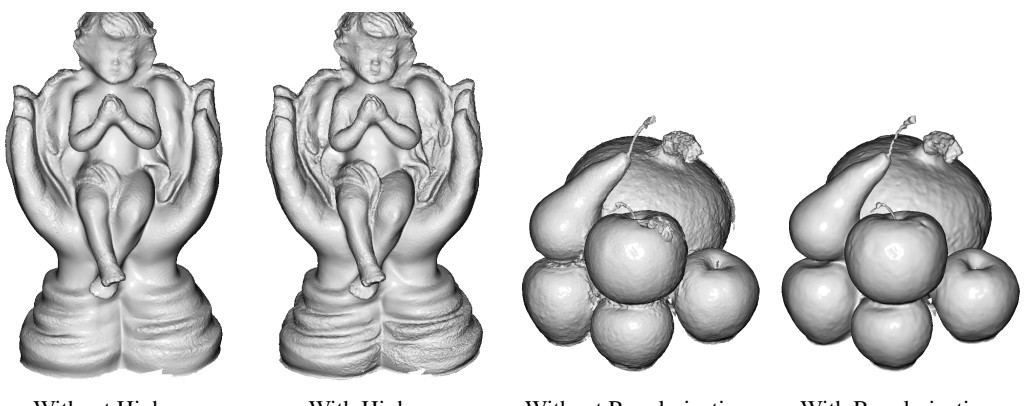

| Without High-res | With High-res | Without Regularization | With Regularization |

Figure 9: Ablation study of Neuralwarp+Ours on the DTU dataset.

## A.5 RESULTS ON NORMAL CONSISTENCY

We report the normal consistency metric in Table 6. This score is obtained by first calculating the mean absolute dot product of the normals in the reconstructed mesh and the normals at the corresponding nearest neighbors in the ground-truth mesh, and then calculating that in reverse.

| Method | 24 | 37 | 40 | 55 | 63 | 65 | 69 | 83 | 97 | 105 | 106 | 110 | 114 | 118 | 122 | Mean |
|---|---|---|---|---|---|---|---|---|---|---|---|---|---|---|---|---|
| NeuS | 0.895 | 0.887 | 0.928 | 0.955 | 0.935 | 0.947 | 0.819 | 0.822 | 0.846 | 0.816 | 0.949 | 0.886 | 0.942 | 0.947 | 0.918 | 0.899 |
| NeuralWarp | 0.917 | 0.908 | 0.964 | 0.959 | 0.931 | 0.944 | 0.898 | 0.826 | 0.840 | 0.825 | 0.947 | 0.896 | 0.942 | 0.942 | 0.940 | 0.912 |
| NeuS+Ours | 0.942 | 0.895 | 0.968 | 0.957 | 0.936 | 0.953 | 0.906 | 0.817 | 0.856 | 0.817 | 0.956 | 0.897 | 0.943 | 0.951 | 0.927 | 0.915 |
| NeuralWarp+Ours | 0.920 | 0.909 | 0.968 | 0.955 | 0.947 | 0.944 | 0.910 | 0.829 | 0.852 | 0.822 | 0.953 | 0.901 | 0.940 | 0.943 | 0.938 | 0.915 |

Table 6: Quantitative results for the normal consistency metric on the DTU dataset.

## A.6 COMPARISON TO PLENOXELS AND INSTANT-NGP.

To better see the difference between our method and Plenoxels or Instant-NGP, we perform some detailed experiments on the Scan-24 of DTU dataset Jensen et al. (2014).

The first one is the original version of Instant-NGP with hash tables of size $2^{24}$. The batch size (the number of sampled points in one iteration) is set to 65536 for a fair comparison of the efficiency. Other parameters are kept the same as the original code, while the images and camera poses are kept the same as in our experiments. The result of this experiment can be seen in Figure 11, where the accuracy of novel view synthesis is high but the surface is noisy.

The second one is NeuS+Hash, where we add to NeuS with the hash tables of the same settings as the Instant-NGP experiment. We add the feature embedding from the hash tables to NeuS while keeping other parameters of NeuS unchanged. The result of this experiment can be seen in Figure 11, where NeuS+Hash could obtain the surface with high quality in most regions, but there are still some defects on the surface, especially in the less-seen regions, which take a weak place in the hash collision.

The third one is NeuS+Plenoxels, where we add to NeuS with a single-volume encoding of resolution $256^3$. We still keep 2 layers of MLPs, because the result is a mess without any MLPs. We

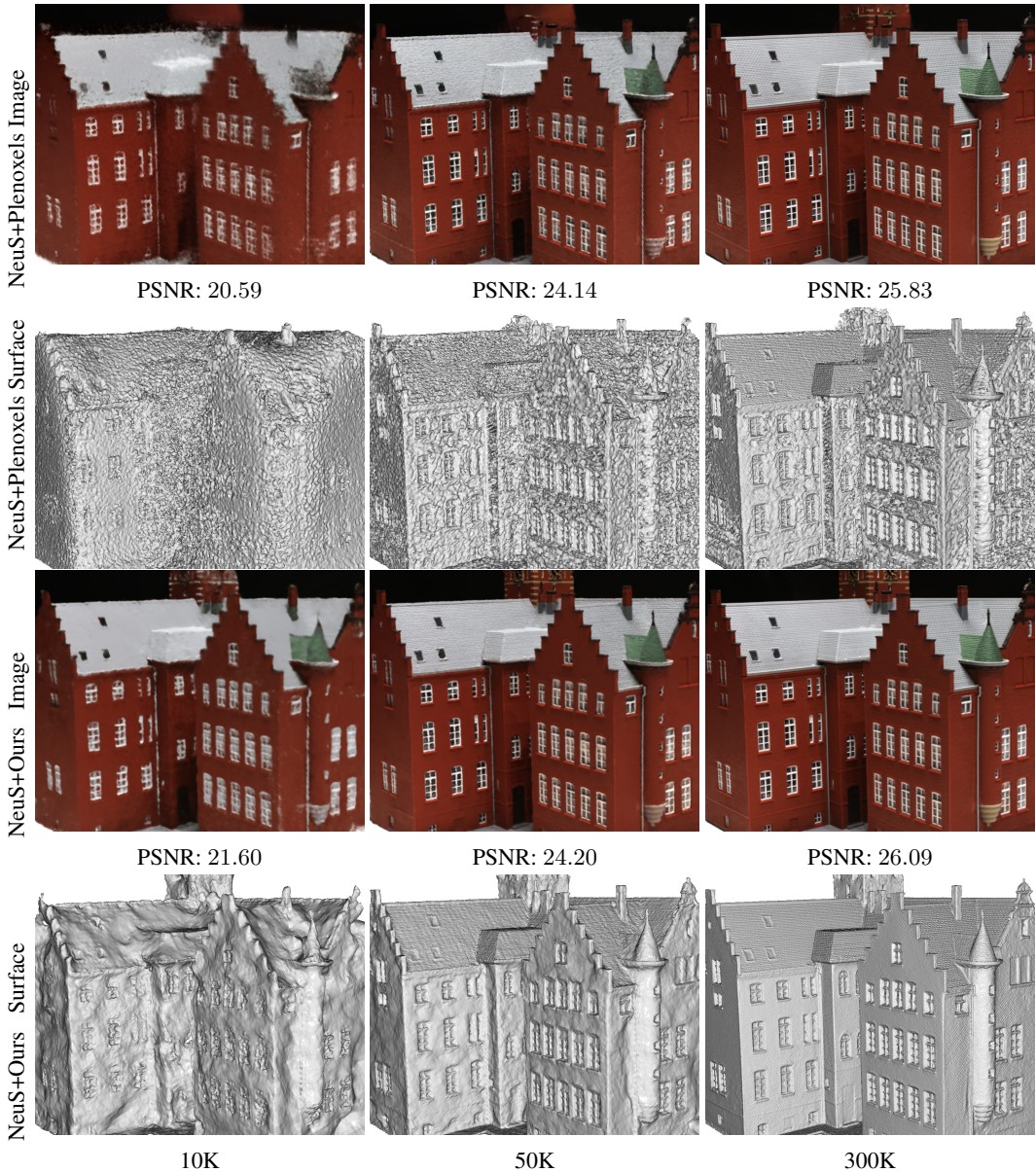

Figure 10: Visual comparison to Plenoxels on DTU scan-24. The rendered novel view and the extracted mesh are displayed with respect to the iteration number.

also tried the original version of Plenoxels, which performs badly on this dataset. From the results displayed in Figure 10, the results of NeuS+Plenoxels is noisy, despite that the image quality is high.

However, the convergence speed of Instant-NGP is faster than ours. This is because we still use large MLPs equipped in NeuS, which requires more time to optimize. Also, the hash tables are memory efficient, such that our method requires about three times more memory compared to Instant-NGP or NeuS+Hash in these experiments.

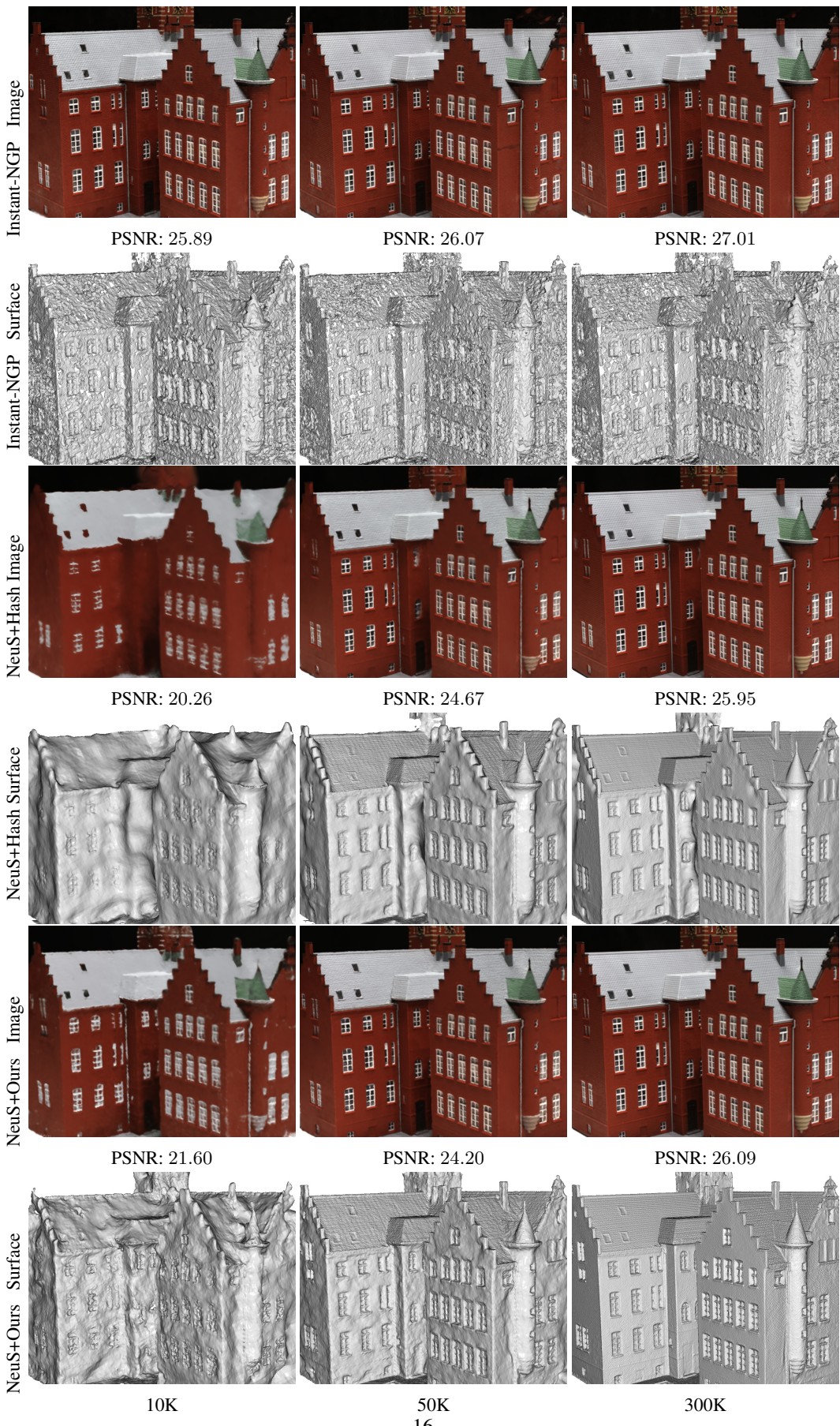

Figure 11: Visual comparison to Instant-NGP on DTU scan-24. The rendered novel view and the extracted mesh are displayed with respect to the iteration number.

## A.7 Evaluation of novel view synthesis

To evaluate the effectiveness of our method in novel view synthesis, we perform the experiments and report the results in Table 7, Table 8, and Figure 12. We select one of each 7 images from the original image set of DTU as the test views, and the remaining images as the training views. The accuracy of novel view synthesis on 15 scans of DTU is reported in Table 7, while the PSNR result with respect to iteration number is reported in Table 8. From the results in these two tables and the images displayed in Figure 12, we can see both the accuracy and convergence speed of the novel view synthesis benefit from our hierarchical volume encoding.

| Method | 24 | 37 | 40 | 55 | 63 | 65 | 69 | 83 | 97 | 105 | 106 | 110 | 114 | 118 | 122 | Mean |
|---|---|---|---|---|---|---|---|---|---|---|---|---|---|---|---|---|
| NeuS | 24.54 | 24.67 | 25.04 | 26.83 | 27.97 | 29.34 | 17.90 | 27.2 | 25.22 | 16.3 | 31.71 | 31.44 | 28.86 | 33.14 | 21.60 | 26.12 |
| NeuS+Ours | 26.00 | 24.98 | 26.76 | 26.94 | 27.06 | 30.08 | 28.44 | 27.1 | 25.53 | 27.68 | 32.37 | 31.81 | 29.73 | 33.70 | 34.45 | 28.84 |

Table 7: PSNR results for the novel view synthesis on the test images of DTU dataset.

| Method | 5K | 10K | 20K | 30K | 50K | 100K | 200K | 300K |
|---|---|---|---|---|---|---|---|---|
| NeuS | 18.22 | 19.01 | 20.38 | 20.88 | 21.49 | 22.25 | 23.84 | 24.54 |
| NeuS+Ours | 20.03 | 21.87 | 23.04 | 23.67 | 23.68 | 24.84 | 25.84 | 26.00 |

Table 8: PSNR results for the novel view synthesis with respect to the iteration number on 7 test images of DTU scan-24.

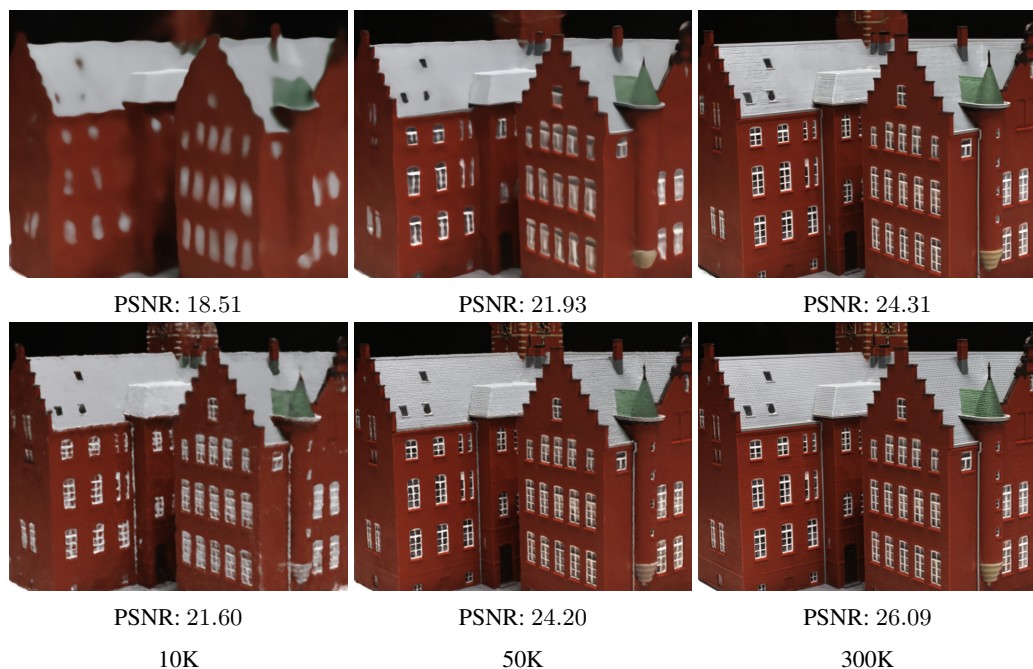

| PSNR: 18.51 | PSNR: 21.93 | PSNR: 24.31 |

| PSNR: 21.60 | PSNR: 24.20 | PSNR: 26.09 |

| 10K | 50K | 300K |

Figure 12: Novel view synthesis on DTU scan-24. The top row is the results of NeuS while the bottom row is the results of NeuS+Ours.

## A.8 Visualization of volume encodings

To inspect how the volumes help encode the 3D space, we visualize the volumes in Figure 13. Due to the 4D structure of the volumes, we perform the marching cubes with a threshold to do the visualization. Following the threshold selection of NeRF, the threshold is set to 0 since the values of most of the voxels are distributed around 0, as shown in the histogram of Figure 13. From the displayed histograms and volumes, we can see the values in the volumes are set to be distributed randomly ($\mu = 0, \sigma = 0.02$) before the training, and distributed close to the object surface after the training. Also, the volumes in different resolutions represent different fineness of the scene.

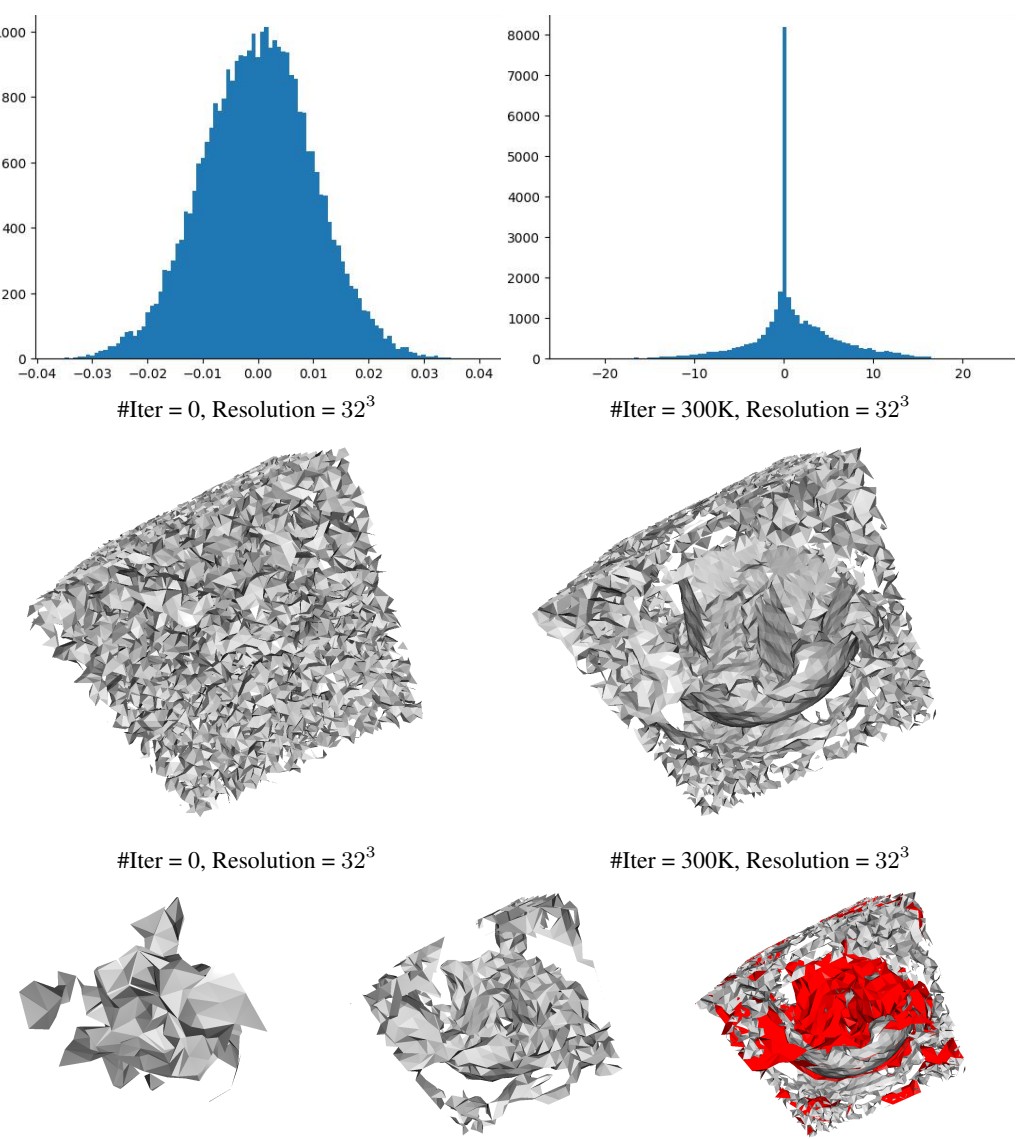

Figure 13: Visualization of the volume encodings. The top row presents the histograms of the values of the voxels in the volume with the resolution of $32^3$, of which the first one is the initialized volume before the training, and the second one is the final volume after the training. The corresponding meshes are displayed in the second row, and more meshes are displayed in the third row.

