# OpenReview forum: "HIVE: HIerarchical Volume Encoding for Neural Implicit Surface Reconstruction"
_ICLR.cc/2023/Conference — Submitted to ICLR 2023_

### Official Review · Reviewer_ZvSi · 2022-10-23

**Confidence:** 5
**Correctness:** 3
**Technical Novelty And Significance:** 3
**Empirical Novelty And Significance:** 4
**Recommendation:** 6

**Clarity, Quality, Novelty And Reproducibility:**

Clarity: some of the claims in the paper is not properly addressed of proofed by experiments.

Reproducibility: The idea of the paper is clearly state and should be easy for readers to reproduce.

**Strength And Weaknesses:**

Strength

- The proposed strategy can be widely adopted in different coordinate-based neural implicit surface papers.

- The underlying technique of this paper is very simple and easy to understand, yet it significantly improves the surface reconstruction quality of the previous state-of-the-art methods in various public available datasets.



Weaknesses

Please explain more on the claims in the paper :
- In page 4, Sec 3.3, "This works well for novel view rendering but has problems in surface reconstruction, because the simple MLP architecture makes learning and optimization inefficient".
Why a simple MLP makes the learning inefficient? Why is it inefficient in geometry learning, but good at RGB learning?

- In page 5, on top: "This feature volume can naturally encode the knowledge about the 3D space of the object".
Why is a randomly initialized volume encoding can "naturally" encode the knowledge about an object? How is this volume encoding looks like when fully trained? More visualization would be helpful here in revealing the use of such encoding.


Additional experiments suggestions:

I am curious to see whether the proposed method can improve the RGB reconstruction or not? It will be good to showcase some novel view synthesis evaluations. And it would be good to explain more on why this encoding can or cannot help RGB reconstruction.

Typos:
In page 4, first line, "with which another implicit",

**Summary Of The Paper:**

This paper proposed a multi-scale learnable feature volume encoding for neural implicit surface reconstruction.
The main idea of the paper is to train a neural implicit surface function from coarse to fine in 3 stages. In each stage, a learnable volume feature encoding is concatenated to the input of the implicit function. The authors also proposed a volume sparsify procedure to discard the empty voxel for memory efficiency.
The proposed strategy can be widely adopted in different coordinate-based neural implicit surface papers.

The underlying technique of this paper is very simple and easy to understand, yet it significantly improves the surface reconstruction quality of the previous state-of-the-art methods in various public available datasets.

**Summary Of The Review:**


I found the idea of this paper very simple yet very effective.
The proposed method can also be easily applied to other works. The paper achieves state-of-the-art performance on surface reconstruction.
But some claims in the paper is not properly addressed and nor proofed via experiments. Please see the weakness.

I think the paper at least provides good empirical results on utilizing volume encoding in improving SDF surface reconstruction. On the other hand, it would be good if the authors could also provide more insights on how such encoding assists the geometry reconstruction, and whether such encoding can be applied to broader use (such as novel synthesis).

---

> ### Author Response · Authors · 2022-11-18
> **Author Responses to Reviewer ZvSi**
>
> Thanks for the constructive comments and critical assessment of our manuscript.
>
> 1. Why a simple MLP makes the learning inefficient? Why is it inefficient in geometry learning, but good at RGB learning?
>
>     We have removed this sentence.
>     According to previous papers, MLPs may cause difficulties in optimizing the target shape as is observed in mesh and point-cloud processing[1, 2].
>
>
> 2. Visualization of volume encoding.
>
>     To inspect how the volumes help encode the 3D space, we **visualize the volumes in Figure 13 in the modified manuscript**.
>     Following the threshold selection of NeRF, the threshold is set to $0$ since the values of most of the voxels are distributed around $0$,  as shown in the histogram of Figure 13.
>     From the displayed histograms and volumes, we can see the values in the volumes are set to be distributed randomly ($\mu = 0, \sigma = 0.02$) before the training, and distributed close to the object surface after the training.
>     Also, the volumes in different resolutions represent different fineness of the scene.
>
> 3. Novel view synthesis evaluations.
>
>     We present the evaluation about the novel view synthesis in the below tables and **Figure 12 in the modified manuscript**.
>     We select one of each 7 images from the original image set of DTU as the test views, and the remaining images as the training views.
>
>     The accuracy of novel view synthesis on 15 scans of DTU is reported in Table 1, while the PSNR result with respect to iteration number is reported in Table 2.
>     From the results in these two tables, we can see both the accuracy and convergence speed of the novel view synthesis benefit from our hierarchical volume encoding.
>
>
>     |Method|24|37|40|55|63|65|69|83|97|105|106|110|114|118|122|Mean|
>     |:--:|:--:|:--:|:--:|:--:|:--:|:--:|:--:|:--:|:--:|:--:|:--:|:--:|:--:|:--:|:--:|:--:|
>     |NeuS|24.54|24.67|25.04|26.83|27.97|29.34|17.90|27.2|25.22|16.3|31.71|31.44|28.86|33.14|21.60|26.12|0.899|
>     |NeuS+Ours|26.00|24.98|26.76|26.94|27.06|30.08|28.44|27.1|25.53|27.68|32.37|31.81|29.73|33.70|34.45|28.84|
>
>     Table 1. PSNR results for the novel view synthesis on the test images of DTU dataset.
>
>
>     |Method|5K|10K|20K|30K|50K|100K|200K|300K|
>     |:--:|:--:|:--:|:--:|:--:|:--:|:--:|:--:|:--:|
>     |NeuS|18.22|19.01|20.38|20.88|21.49|22.25|23.84|24.54|
>     |NeuS+Ours|20.03|21.87|23.04|23.67|23.68|24.84|25.84|26.00|
>
>     Table 2. PSNR results for the novel view synthesis with respect to the iteration number on 7 test images of DTU scan-24.
>
>
> [1] Julian Chibane, Thiemo Alldieck, and Gerard Pons-Moll. Implicit functions in feature space for 3d shape reconstruction and completion.
>
> [2] Songyou Peng, Michael Niemeyer, Lars Mescheder, Marc Pollefeys, and Andreas Geiger. Convolutional occupancy networks.

---

### Official Review · Reviewer_wTJb · 2022-10-24

**Confidence:** 4
**Correctness:** 3
**Technical Novelty And Significance:** 2
**Empirical Novelty And Significance:** 3
**Recommendation:** 3

**Clarity, Quality, Novelty And Reproducibility:**

The submitted manuscript is overall of good quality, but lacks clarity on a few specific points. One is the treatment of Instant NGP as described above, which leaves out a lot of detail such that it would be impossible to reproduce the experiment. A few statements that are critical of prior approaches to neural reconstruction are not clear or concrete on what the issues being pinpointed are. For example, in section 3.3 the manuscript states that MLP-based renderings have “problems in surface reconstruction, because the simple MLP architecture makes learning and optimization inefficient,” but it is not clear what is meant by a “simple” MLP, what is the distinction between “learning” and “optimization,” or what kind of efficiency is lacking. Similarly, in the related work the manuscript states that when using NeRF to represent a scene, “the surface extracted from the implicit network usually has some defect,” but it is not clear which defects are meant or which surface extraction method is being reference, as NeRF does not represent surfaces and thus does not provide an unambiguous method of surface extraction.

**Strength And Weaknesses:**

The paper identified a valid issue with many neural scene representations, i.e. the inductive bias towards lower-frequency radiance, density, and/or signed distance fields which often results in overly-smooth renderings. The paper also proposes a sensible strategy to mitigate this issue, and show that their technique significantly reduces reconstruction errors when applied to a variety of scene rendering models. The ablation studies provide interesting insights into the value added by the hierarchical volumes and the influence of various hyperparameters.

The key weakness of the paper is that it lacks technical novelty. However, it is not clear from reading the manuscript this is the case. There are important gaps in the related work cited, and the related work that is cited is given a fairly cursory treatment. Instant Neural Graphics Primitives introduces a very similar approach in which a scene is volumetrically encoded, and it can be used interchangeably with the proposed hierarchical volumes. However, Instant-NGP is only mentioned at the very end of the manuscript. An experiment is performed to compare the performance of Instant-NGP vs. HIVE and the results indicate that HIVE leads to superior performance. However, the experiment is not described in detail and as such it is hard to interpret the results. The manuscript mentions an advantage of HIVE over Instant-NGP, namely that it does not experience hash collisions which can impair performance, but does not mention the corresponding advantage of Instant-NGP over HIVE which is that the memory cost can be kept to O(1) in the scene volume whereas HIVE is O(N) in the scene volume (with some scalar multiplier resulting from sparsification). Was the amount of memory used to represent the scenes balanced between the two approaches for the experiment presented in Table 1? The manuscript also provides no runtime analysis of the proposed approach, so we do not know how it compares to Instant-NGP which is known to enable fast inference of scene representations.

The manuscript also completely fails to mention a number of highly related works on hierarchical sparse representation of scenes for neural reconstruction, such as Neural Geometric Level of Detail (Takikawa et al. 2021), PlenOctrees (Yu et al. 2021), and ACORN (Martel et al. 2021). NGLoD in particular introduces an extremely similar scene representation. Unfortunately the two approaches are evaluated on different datasets and thus we do not know which performs better or why.


**Summary Of The Paper:**

This paper describes a series of experiments in which a hierarchical feature volume is used to encode a latent representation of a scene. The volume defines a continuous feature field through linear interpolation of the features associated with discrete voxel locations. The resultant feature at any point in space can then be passed as input to a neural network or networks to render images of the encoded scene. A variety of rendering methods can make use of this same scene representation, such as NeRF, NeuS, etc. The paper also describes a method of progressively optimizing the feature volumes in a coarse-to-fine order, and a sparsification strategy for the higher-resolution volumes to avoid using too much memory.

**Summary Of The Review:**

The manuscript provides new experiments showing that the addition of a sparse hierarchical volume encoding to a neural scene renderer can improve performance of a variety of existing baseline methods. However, the technical novelty over other previously published sparse hierarchical scene representations is very small, and the manuscript not only fails to make a case for the use of the proposed technique over the highly similar existing approaches.

---

> ### Author Response · Authors · 2022-11-18
> **Author Responses to Reviewer wTJb (1/2)**
>
> Thanks for the constructive comments and critical assessment of our manuscript.
>
> 1. Technical novelty and the relationship to existing works.
>
>     Due to being too focused on the task of neural surface reconstruction from multiple views, sorry for the insufficient discussion about the relationship to existing works. We agree that there are already some similar works, but there is still critical differences between our method and others.
>
>     The difference from Plenoxels, DVOG, TensoRF is that there is only a single geometric volume, whereas the first important insight in this paper is that using multiple small-feature-channel hierarchical volumes performs much better than one large-feature-channel volume, while also saving the memory consumption, as shown in Figure.7 in the original paper (Figure.8 in the modified version: ablation study about the number of the volumes). In a single-geometric-volume case, each voxel is updated in isolation. Due to the high degree of freedom in optimizing one voxel, it is hard to maintain a globally coherent shape. But in hierarchical volumes, the voxels in low-resolution volumes contain the information of larger space.
>
>     The difference from Instant-NGP lies in that they use hash tables to store the volume, which may cause some problems due to the hash collision, as stated in their original paper: "However, it also exhibits visually undesirable surface roughness that we attribute to randomly distributed hash collisions." "The graininess is a result of hash collisions that the MLP is unable to fully compensate for."
>
>     Another difference from Instant-NGP lies in the resolution of the hierarchical volumes, which is also another important insight of our paper. As shown in Figure.7 in the original paper (Figure.8 in the modified version:  ablation study about the resolution combination of volume), the resolution of the smallest volume is critical to the final performance, because the low-resolution volumes enforce the spatial consistency to keep the shape globally coherent.
>
>     **From the visualization in Figure 11 in the modified manuscript**, we can also see the defects of Instant-NGP, although the accuracy of its novel view synthesis is high.
>
>
> 2. Details of the comparison experiments to Instant-NGP.
>
>     In the modified manuscript, we present two experiments about Instant-NGP:
>
>     + The first one is the original version of Instant-NGP with hash tables of size $2^{24}$. The batch size (the number of sampled points in one iteration) is set to $65536$ for a fair comparison of the efficiency experiments. Other parameters are kept the same as the original code, while the images and camera poses are kept the same as in our experiments.
>     The result of this experiment can be seen in **Figure 11** in the modified manuscript, where the accuracy of novel view synthesis is high but the surface is noisy.
>
>     + The second one is the NeuS+Hash, where we add to NeuS with the hash tables of the same settings as the Instant-NGP experiment. We add the feature embedding from the hash tables to NeuS while keeping other parameters of NeuS unchanged.
>     The result of this experiment can be seen in **Figure 11** in the modified manuscript, where NeuS+Hash could obtain the surface with high quality in most regions, but there are still some defects on the surface, especially in the less-seen regions, which take a weak place in the hash collision.
>
> 3. Advantage of Instant-NGP, memory and efficiency analysis
>
>     We agree that the memory cost of Instant-NGP can be kept to O(1) in the scene volume, and the speed of Instant-NGP is quite fast.
>
>     As shown in **Figure 11** in the modified manuscript, Instant-NGP could obtain high-quality novel view synthesis with only 10K iterations, while our method requires 100K iterations. This is because there are two large MLPs in our framework (same as NeuS), which require more time to optimize.
>     For memory consumption, our method requires three times more memory compared to Instant-NGP or NeuS+Hash.
>     However, although the view synthesis quality of Instant-NGP is quite high, its surface quality is low with much noise.
>    NeuS+Hash could obtain the surface with high quality in most regions, but there are still some defects on the surface, especially in the less-seen regions, which take a weak place in the hash collision.

---

> > ### Author Response · Authors · 2022-11-18
> > **Author Responses to Reviewer wTJb (2/2)**
> >
> > 4. Related work. Neural Geometric Level of Detail (Takikawa et al. 2021), PlenOctrees (Yu et al. 2021), and ACORN (Martel et al. 2021).
> >
> >     PlenOctrees or Plenoxels use only a single geometric volume in their method. One of our insights is that using multiple small-feature-channel hierarchical volumes performs much better than one large-feature-channel volume, while also saving the memory consumption.
> >
> >     In a single-geometric-volume case, each voxel is updated in isolation. Due to the high degree of freedom in optimizing one voxel, it is hard to maintain a globally coherent shape. But in hierarchical volumes, the voxels in low-resolution volumes contain the information of larger space.
> >
> >     ACORN use MLPs to compute the volumes, and optimize the MLPs in the training, while we directly optimize the volumes.
> >     Neural Geometric Level of Detail sums up the features from a large-feature-channel octree, while we concatenate the features from multiple small-feature-channel volumes, in which case our method consumes less memory.
> >
> >     In addition, ACORN and Neural Geometric Level of Detail are all designed for SDF encoding, where the ground-truth SDF is provided and thus a different task from our work.
> >
> >
> >
> > 5. It is not clear what is meant by a “simple” MLP, what is the distinction between “learning” and “optimization,”. It is not clear which defects are meant or which surface extraction method is being reference, as NeRF does not represent surfaces and thus does not provide an unambiguous method of surface extraction.
> >
> >     "simple" here means the standard MLPs used in NeRF without volumes.
> >
> >     We have changed "learning and optimization" to "optimization" in the modified manuscript.
> >
> >     The surface is extracted by marching cubes. As stated in Unisurf, NeRF "requires careful tuning of the density threshold and leads to artifacts due to the ambiguity present in the density". The defects can be seen from Figure 3 of Unisurf and **Figure 2 and Figure 11 in our modified manuscript**, where the mesh of Instant-NGP contains many defects.
> >
> >
> > [1] Takikawa T, Litalien J, Yin K, et al. Neural geometric level of detail: Real-time rendering with implicit 3D shapes.
> >
> > [2] Martel J N P, Lindell D B, Lin C Z, et al. Acorn: Adaptive coordinate networks for neural scene representation.
> >
> > [3] Fridovich-Keil S, Yu A, Tancik M, et al. Plenoxels: Radiance Fields Without Neural Networks.

---

### Official Review · Reviewer_hdwP · 2022-10-25

**Confidence:** 5
**Correctness:** 4
**Technical Novelty And Significance:** 2
**Empirical Novelty And Significance:** 2
**Recommendation:** 5

**Clarity, Quality, Novelty And Reproducibility:**

The paper is written and described clearly, and it is easy to follow. The proposed concept is simple so it should be easily reproducible. However, I have doubts on the originality and novelty of this work. It is very similar to Plenoxels but only applied to the application of surface reconstruction. There is also not much additional insights on how the proposed HIVE differ from prior works like Plenoxels.

**Strength And Weaknesses:**

Strengths:
+ This paper presents a simple and effective improvement over frequency encoding used in neural surface reconstruction methods like NeuS and VolSDF. Using explicit (hybrid) representations have recently shown success to accelerate convergence and improve view synthesis results [A,B,C]; this paper applies the same concept to surface reconstruction.
+ The proposed HIVE adds significant improvement over baseline models (NeuS and VolSDF) in reconstruction quality.
+ The method description is easy to follow and understand.

Weaknesses:
- The authors have missed very important references that improves NeRF with explicit (hybrid) encodings, listed as [A,B,C]. In particular, Plenoxels [A] have already proposed explicit encoding with a voxel pruning strategy. In some sense, this paper proposed almost exactly the same concept, but only applied to surface reconstruction. In addition, the authors have ignored a large body of prior works on using octrees to represent 3D scenes with hierarchical voxel grids. A survey can be found in [D].
- I think it would be good to have discussions on the drawbacks of the proposed HIVE. One limitation I can see is that HIVE is currently limited to the resolution of 1024^3. Although the higher resolutions can be initialized from sparsified voxels (with an additional dilation step), the amortized memory growth will be N^2 (proportional to the surface area) and eventually it will still be memory bounded. So it doesn't seem like HIVE is much more scalable.
- Another drawback of HIVE is that granular details are not recoverable if certain voxels were pruned at resolution 256^3. So there is a heavy assumption that the coarse shapes should be fully captured at 256^3. This is not a limitation for more naïve frequency encoding from NeRF.
- The improvement on BlendedMVS scenes seems significant over NeuS and somewhat better than NeuralWarp, but the resolution of the recovered surfaces do not seem to be as high resolution as expected (1024^3?). It would be great if the authors could clarify.
- The NeuS+hash results seem questionable. From the results of Muller et al., Instant NGP has even more representation capability than the original NeRF, and artifacts shown in Fig 6(c) are not to be expected. It would be good if the authors could clarify; otherwise I would not think this is a fair and faithful comparison.
- Minor: Fig 6 is hard to understand without sufficient descriptions in the captions.
- Minor: I don't think equations 7 and 8 are necessary; they are just binary dilation and index lookup operations.

[A] Yu et al. "Plenoxels: Radiance Fields without Neural Networks." CVPR 2022
[B] Sun et al. "Direct Voxel Grid Optimization: Super-fast Convergence for Radiance Fields Reconstruction." CVPR 2022
[C] Chen et al. "TensoRF: Tensorial Radiance Fields." ECCV 2022
[D] Knoll. "A survey of octree volume rendering methods." 2006

**Summary Of The Paper:**

This paper presents an improved design of spatial encoding for neural surface reconstruction methods, e.g. NeuS and VolSDF. Instead of using frequency positional encoding as in NeRF, the authors propose to explicitly encode the spatial information with multi-scale voxels storing feature vectors. In particular, the encoding consists of dense voxel grids up to the resolution of 256^3, whereas subsequent higher resolutions (512^3 and 1024^3) are derived from sparsified results from the previous levels. Experiments are focused on the DTU benchmark with additional visualizations on BlendedMVS and EPBL, showing improvements over frequency encoding.

**Summary Of The Review:**

I think this is a nice paper that presents surface reconstruction results that have shown observable improvements, but I have major concerns on the novelty of this work. While I understand novelty is not the sole evaluation criterion, I think there are insufficient discussions on (a) how this work differ from the others, (b) deeper insights on the technical parts of the methods, (c) limitations of the proposed method. I am thus leaning towards a reject at this point.

---

> ### Author Response · Authors · 2022-11-18
> **Author Responses to Reviewer hdwP**
>
> Thanks for the constructive comments and critical assessment of our manuscript.
>
> 1. Difference from Plenoxels, DVOG, TensoRF.
>
>     Sorry for missing the discussion about our difference from existing methods.
>
>     Volume encoding has already been proposed in Plenoxels, DVOG, and TensoRF, but there is only a single geometric volume in their method. One of our insights is that using multiple small-feature-channel hierarchical volumes(e.g. {$2^{3}\times4$, $4^{3}\times4$,$8^{3}\times4$,..,$256^{3}\times4$}) performs better than one large-feature-channel volume(e.g. $256^{3}\times32$), while also saving the memory consumption(the amount of optimized parameters is about 1/8), as shown in Figure 7 in the original paper (**Figure 8 in the modified version: ablation study about the number of the volumes, the related description is in section 4.3: Setup of embedding volumes**).
>
>     In a single-geometric-volume case, each voxel is updated in isolation. Due to the high degree of freedom in optimizing one voxel, it is hard to maintain a globally coherent shape. But in hierarchical volumes, the voxels in low-resolution volumes contain the information of larger space, which could enforce spatial consistency to keep the shape smooth.
>
>     We have added this discussion in the modified manuscript, and also **added the comparison experiment in Figure 10 (applied Plenoxels to surface reconstruction)**, from where we can see the reconstructed surface is noisy with only one high-resolution volume.
>
>
> 2. Limitation of memory.
>
>     Yes, our framework is memory bounded and has difficulties in reconstructing large-scale scenes. This is also a common problem of volume-based methods. We have added this limitation discussion to the appendix in the modified manuscript.
>
> 3. Limitation of sparsify.
>
>     We agree that the voxels pruned at low resolution cannot be recovered at high resolution. To address this, we dilate the low-resolution surface, such that usually all the useful voxels would be included. Only in a few cases, the voxels would be pruned by mistake.
>     We have added this limitation discussion to the appendix in the modified manuscript.
>
> 4. Resolution of the displayed meshes.
>
>     Sorry for the unclear statement. For a fair comparison, the resolution of the marching cubes is set to 512 for all methods, although the volume resolution of our method is 1024.
>     We have added this explanation in the modified manuscript.
>
> 5. The performance of NeuS+Hash.
>
>     We agree that Instant NGP has strong representation capability, but it performs better in novel view synthesis than surface reconstruction.
>     As stated in the original paper of Instant-NGP, the hash collisions do cause some problems: "However, it also exhibits visually undesirable surface roughness that we attribute to randomly distributed hash collisions." "The graininess is a result of hash collisions that the MLP is unable to fully compensate for."
>     This hash collision problem is more serious in surface reconstruction since the collision in view synthesis could be better compensated for by MLPs.
>
>     In the modified manuscript, we present two experiments about Instant-NGP:
>
>     + The first one is the original version of Instant-NGP with hash tables of size $2^{24}$. The batch size (the number of sampled points in one iteration) is set to $65536$ for a fair comparison of the efficiency experiments. Other parameters are kept the same as the original code, while the images and camera poses are kept the same as in our experiments.
>     **The result of this experiment can be seen in Figure 11** in the modified manuscript, where the accuracy of novel view synthesis is high but the surface is noisy.
>
>     + The second one is the NeuS+Hash, where we add to NeuS with the hash tables of the same settings as the Instant-NGP experiment. We add the feature embedding from the hash tables to NeuS while keeping other parameters of NeuS unchanged.
>     **The result of this experiment can be seen in Figure 11** in the modified manuscript, where NeuS+Hash could obtain the surface with high quality in most regions, but there are still some defects on the surface, especially in the less-seen regions, which take a weak place in the hash collision.
>
> 6. Minor questions: Caption of Fig 6; Removing equations 7 and 8.
>
>     We have added more explanation to the caption of Figure 6 (Figure 7 in the modified version), and have removed equations 7 and 8.

---

### Official Review · Reviewer_frLo · 2022-10-25

**Confidence:** 3
**Correctness:** 4
**Technical Novelty And Significance:** 2
**Empirical Novelty And Significance:** 3
**Recommendation:** 6

**Clarity, Quality, Novelty And Reproducibility:**

- Clarity & Quality

The manuscript is overall easy to understand with nice figures. Although some parts of manuscript, such as the third contribution in page 2 should be modified to more concrete and specific sentence. Also, one could add captions on Table 1 & 2 explicitly stating the table is about Chamfer distance.

- Novelty

As stated in the weakness section, although using hierarchical feature volume encoding is not new, I enjoy the simplicity.

- Reproducibility

The authors promised to release the code.


**Strength And Weaknesses:**

**Strength**

- Performance

The work shows impressive performance both quantitatively and qualitatively. I was impressed by figure1 & 5 reconstructing high frequency details from images.

- Simple and efficient scheme

Using hierarchical features for implicit reconstruction makes sense and is straightforward. I also enjoyed a simple idea of obtaining sparse high resolution volume by using multi-stage optimization and dilating the low resolution reconstructed implicit function.


**Weakness**

- Using hierachical feature volume encoding has been used in the field of computer vision for a long time with various tasks, including surface reconstuction, such as [1].

This idea is not brand new, however, I did enjoy the performance increase with respect to the simplicity of the method.


- The work can be strengthened by adding more experiments for NeuralWarp + HIVE.

The authors have reported all scores for both Neus and NeuralWarp with and without HIVE for table 1 & 2. I think adding restuls for NeuralWarp + HIVE in Figure 6 and ablation study can further strengthen the paper, because this paper shows enhancement of the results based on previous works. Because this work is an add on to other methods, I believe these results are necessary to show the performance increase.

- The work can be strengthened by adding more quantitative metric than Chamfer distance.

Table 1, 2 shows the Chamfer distance compared to other methods. While the Chamfer is a good metric to compare the similarity of to points clouds, there has been many questions [2] whether the Chamfer distance could capture the high frequency details or overall ditributional similarity between two point clouds. I would suggest adding normal consistency as in [3] or some other metrics that show that the method can capture the high frequency surfaces.

[1] Chibane et al. Implicit functions in feature space for 3d shape reconstruction and completion. CVPR, 2020.

[2] Wu et al. Density-aware Chamfer Distance as a Comprehensive Metric for Point Cloud Completion. NeurIPS, 2021

[3] Mescheder et al. Occupancy Networks: Learning 3D Reconstruction in Function Space. CVPR 2019


**Summary Of The Paper:**

The work tackles on reconstructing the implicit surface from images. The main contribution of the work is to propose a hierarchical volume encoding for neural implicit surface reconstruction, which can be easily be plug-and-played on top of other existing works. Although the hierarchical volume encoding scheme has been widely used in computer vision community, experiments on widely used datasets, such as DTU, BlendedMVS show that the work clearly enhances other methods.

**Summary Of The Review:**

Although the proposed hierarchical volume encoding itself is not novel in the field of surface reconstruction, I believe that the pros of having better performance while being simple is very practical and will be beneficial to community if the additional figures and experiments are done as stated in the weakness section. I would like to hear what other reviewers before making the final decision.

---

> ### Author Response · Authors · 2022-11-18
> **Author Responses to Reviewer  frLo**
>
> Thanks for the constructive comments and critical assessment of our manuscript.
>
> 1. Novelty
>
>     Thanks for the appreciation for the simplicity of our method. We believe a simple yet effective method would benefit many other researchers in the community.
>
>     The difference between our method and [1] is that they use convolutional neural networks to get the feature volumes, and optimize the parameters of 3D CNNs in the training. We also tried this method, but found that this is extremely slow since the computing of CNNs is performed on windows while the computing of MLPs is performed on points. Also, in our experiments, optimizing CNNs did not outperform directly optimizing the volumes.
>
> 2. More experiments for NeuralWarp + HIVE.
>
>     We have presented more experiments about NeuralWarp+Ours in the below table and **Figure 9 in the modified manuscript**, from where the effectiveness of our proposed modules are demonstrated.
>
>
>     |Hierarchical|Regular|Sparse |Mean|
>     |:----:|:-----:|:-----:|:--:|
>     |&cross;|&cross;|&cross;|0.68|
>     |&#10004;| &cross;| &cross; |0.64|
>     |&#10004; | &#10004;| &cross; |0.63|
>     |&#10004; | &#10004;| &#10004; |0.61|
>
>     Table 1. Ablation study of NeuralWarp+Ours on the DTU dataset.
>
>
> 3. More quantitative metrics, e.g., normal consistency.
>
>     We report the normal consistency metric in the below table.
>     This score is obtained by first calculating the mean absolute dot product of the normals in the reconstructed mesh and the normals at the corresponding nearest neighbors in the ground-truth mesh, and then calculating that in reverse. We have added this result to the appendix in the modified manuscript.
>
>     |Method|24|37|40|55|63|65|69|83|97|105|106|110|114|118|122|Mean|
>     |:--:|:--:|:--:|:--:|:--:|:--:|:--:|:--:|:--:|:--:|:--:|:--:|:--:|:--:|:--:|:--:|:--:|
>     |NeuS|0.895|0.887|0.928|0.955|0.935|0.947|0.819|0.822|0.846|0.816|0.949|0.886|0.942|0.947|0.918|0.899|
>     |NeuralWarp|0.917|0.908|0.964|0.959|0.931|0.944|0.898|0.826|0.840|0.825|0.947|0.896|0.942|0.942|0.940|0.912|
>     |NeuS+Ours|0.942|0.895|0.968|0.957|0.936|0.953|0.906|0.817|0.856|0.817|0.956|0.897|0.943|0.951|0.927|0.915|
>     |NeuralWarp+Ours|0.920|0.909|0.968|0.955|0.947|0.944|0.910|0.829|0.852|0.822|0.953|0.901|0.940|0.943|0.938|0.915|
>
>
>     Table 2. Quantitative results for the normal consistency metric on the DTU dataset.
>
> [1] Chibane et al. Implicit functions in feature space for 3d shape reconstruction and completion. CVPR, 2020.

---

> > ### Comment · Reviewer_frLo · 2022-12-05
> > **After rebuttal**
> >
> > I appreciate the author's response and the hard work for the additional experiments.
> > The paper seems to be stronger with it.
> >
> > I agree with the other reviewers that the method of extracting hierarchical features is already being widely used in the community.
> > However, I really enjoy the performance considering the simplicity of the method, especially the qualitative examples, and am willing to retain my score.

---

### Author Response · Authors · 2022-11-18
**Paper Revision**

We thank all reviewers for their valuable feedback and constructive comments. We have revised the paper as suggested by the reviewers, and summarize the major revisions as follows:

- More discussion and comparison experiments to existing works about explicit (hybrid) encodings, including Plenoxels and Instant-NGP, in Section 1, Section 4, and Appendix A.6.

- More ablation study on different frameworks as a plug-in: NeuralWarp + HIVE, in Appendix A.4.

- More quantitative metrics than Chamfer distance (the normal consistency which captures the high-frequency details), in Appendix A.5

- Evaluations on novel view synthesis in Appendix A.7.

- More discussion about the motivation and technical novelty in Section 1.

- More description of the relationship to previous works in Section 2.

- Visualization of volume encodings in Appendix A.8.

- Some minor problems.

The other concerns have also been addressed individually in the Author Responses.

---

### Decision · Program_Chairs · 2023-01-20

**Decision:**

Reject

**Justification For Why Not Higher Score:**

Most reviewers have doubts about the originality and novelty of this work. It is very similar to Plenoxels but only applied to the application of surface reconstruction.


**Justification For Why Not Lower Score:**

N/A

**Metareview: Summary, Strengths And Weaknesses:**

This paper was reviewed by four experts in the field. The reviewers raised many concerns regarding the paper: 1) The technical novelty over other previously published sparse hierarchical scene representations (e.g. Plenoxels, Instant NGP) is marginal. 2) Several experiments (e.g. NeuS+Hash) should be improved to be considered fair comparison. Considering the reviewers' concerns, we regret that the paper cannot be recommended for acceptance at this time. The authors are encouraged to consider the reviewers' comments when revising the paper for submission elsewhere.

**Summary Of Ac-Reviewer Meeting:**

N/A